# COMMUNICATION-EFFICIENT FEDERATED LEARNING VIA GRADIENT DISTILLATION

## ABSTRACT

Federated learning revolutionizes collaborative model training across decentralized edge devices, ensuring privacy by avoiding direct data sharing. However, the frequent exchange of model updates introduces a significant communication overhead. The conventional FL process involves transmitting the differences in parameters between old and new models, resulting in redundant gradient communications due to the intricate interplay between model parameters and network architecture. Even minor adjustments to parameters necessitate the retransmission of entire models. In this paper, we introduce a groundbreaking concept known as *gradient distillation*, which decouples model parameters from network architecture, enabling the transmission of only essential information needed for synchronization. By leveraging gradient distillation, we approximate gradient disparities into a synthetic tensor sequence, allowing the recipient to reconstruct the sender's intended model update. This approach eliminates the need to transmit the entire set of raw parameter differences, offering a highly promising solution for achieving greater communication efficiency while without significant accuracy degradation. Experimental results demonstrate that our approach achieves an unprecedented level of gradient compression performance, surpassing widely recognized baselines by an impressive margin of orders of magnitude.

## 1  INTRODUCTION

Federated learning (FL) (McMahan et al., 2017; Shokri & Shmatikov, 2015) is a promising paradigm in machine learning that addresses the challenge of training models on decentralized data sources. Traditional machine learning approaches rely on centralized servers to collect and process all the data used for training. However, in many real-world scenarios, this centralized approach is impractical due to privacy concerns (Ching et al., 2018), regulatory constraints (GDPR, 2016), or the sheer volume of data generated at the edge (Zhou et al., 2019). FL emerged as a solution to this problem by allowing model training to occur directly on edge devices where the data are generated, without the need to transmit sensitive information to a central server. This approach has gained significant traction in recent years, particularly with the proliferation of mobile devices (Wang et al., 2020; Chen et al., 2023), IoT devices (Imteaj et al., 2021; Zhang et al., 2022), and edge computing (Wang et al., 2019; Nguyen et al., 2021). The decentralized nature of FL makes it well-suited for scenarios where data privacy and regulatory compliance are critical, such as in healthcare applications (Xu et al., 2021; Courtiol et al., 2019), financial transactions (Long et al., 2021; Kaplan, 1989), and other contexts where sensitive information is involved (Niu & Deng, 2022; Li et al., 2021).

The inherent design of FL ensures that raw data remains securely stored on edge devices, rendering it inaccessible to the central server. This property is fundamental to the privacy-preserving aspect of FL. Instead of raw data, model updates are transmitted from participating devices to a central server, where they are aggregated to refine the global model (McMahan et al., 2017). Nevertheless, contemporary neural network models are characterized by a staggering number of parameters, often ranging in the millions or even billions. This is particularly exemplified by the immensely popular large language models (Brown et al., 2020), which can possess hundreds of billions of parameters. In many real-world scenarios, especially those involving decentralized data sources like mobile devices, transmitting such vast amounts of model parameters to a central server may be impractical or infeasible due to limitations in network bandwidth, high latency, or intermittent connectivity. The substantial communication cost acts as a barrier, impeding FL from effectively scaling the

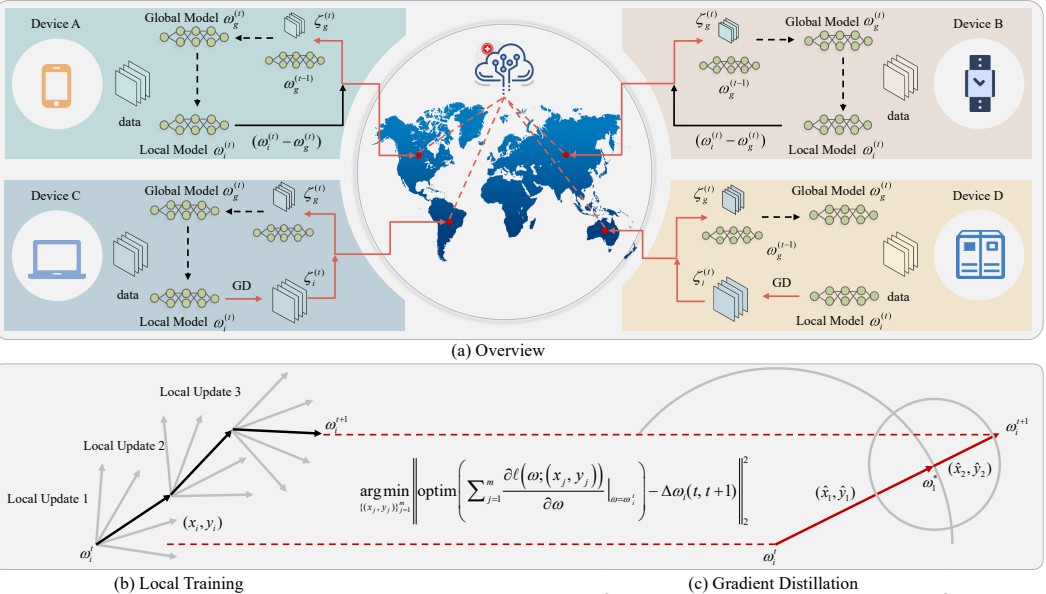

(a) Overview

(b) Local Training  (c) Gradient Distillation

Figure 1: Overview of gradient distillation based FL. (a) Depicts a scenario with four devices, where two resource-constrained devices do not perform local distillation. They solely receive distilled tensors from the server during global model downloads, benefiting from communication savings in the downstream direction. The other two clients engage in local gradient distillation when uploading updates, achieving bidirectional communication efficiency. (b) Shows the process for local model updating. (c) Demonstrates the principle of gradient distillation.

training process to accommodate a larger number of participants. Considering the communication-efficiency issue in FL is imperative for ensuring the practicality, scalability, and energy efficiency of the approach, especially in real-world applications where decentralized data sources are prevalent.

Communication-efficient federated learning has gained significant attention in recent years (Aji & Heafield, 2017; Reisizadeh et al., 2020; Hönig et al., 2022; Liu et al., 2023). Existing approaches can be broadly categorized into several main strategies, each with its own set of limitations. The first strategy involves allowing devices to perform multiple local updates before transmitting their model updates to the central server, thereby reducing the frequency of communication rounds (McMahan et al., 2017; Haddadpour et al., 2019). While this approach reduces the overall transmission data amount, it may result in slower convergence and potential overfitting if not carefully managed. The second strategy focuses on compressing gradient information (Liu et al., 2023; Reisizadeh et al., 2020). This includes methods such as quantization (Hönig et al., 2022), which reduces the precision of gradient values to decrease the transmitted information volume, and sparsification (Aji & Heafield, 2017; Dai et al., 2022), which involves sending only a subset of gradients by retaining the most significant ones and setting others to zero. However, aggressive quantization and sparsification can lead to information loss and potentially hinder model accuracy. Additionally, pruning (Zhu et al., 2022; Wang et al., 2022) is employed to remove less influential weights or neurons from the model, effectively reducing the parameter count and, subsequently, the size of gradients. However, pruning methods require careful hyperparameter tuning and may result in model degradation if not applied judiciously. Another strategy focuses on uploading logits (Sattler et al., 2020) or dataset representations (Xiong et al., 2023). Instead of transmitting raw gradients, devices calculate and send logits (pre-softmax outputs) for their data samples to the central server. Alternatively, devices can convey aggregated representations of their datasets, such as centroids or other statistical summaries (Liu et al., 2022), which serve as a compact proxy for the raw gradients. These strategies offer diverse avenues to tackle communication overhead in FL, each with distinct trade-offs in terms of communication reduction, computational complexity, and potential impact on model performance.

In this paper, we introduce a groundbreaking concept called *gradient distillation*, which exhibits unprecedented performance in FL communication compression, surpassing the renowned baseline FedAvg by a remarkable margin of $1904\times$ on benchmarking medical dataset PathMNIST. The conventional FL technique involves transmitting parameter differences between old and new models,

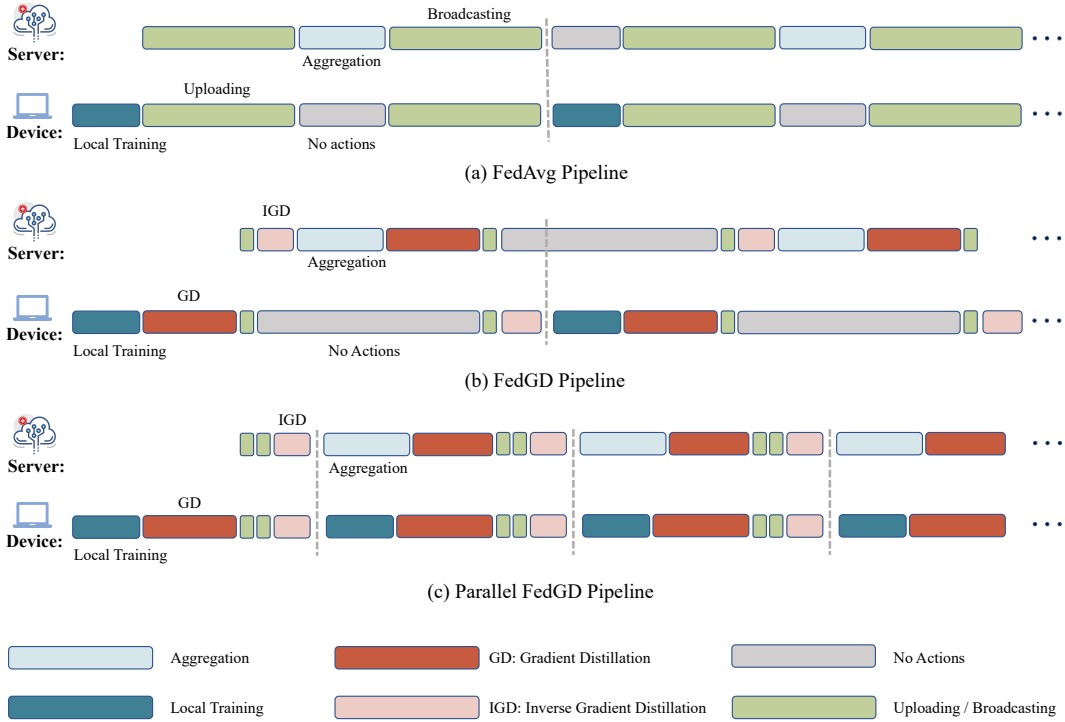

Figure 2: The pipeline of FedAvg, FedGD, and Parallel FedGD.

resulting in redundant gradient communication due to the intricate relationship between model parameters and network architecture. Even minor parameter adjustments necessitate the retransmission of entire models. To tackle this issue, we propose decoupling model parameters from network architecture, enabling transmission of only essential information for synchronization. By employing gradient distillation, we approximate gradient disparities into a synthetic tensor sequence, allowing the recipient to reconstruct the sender's intended model update. This parameter-structure decoupling leads to a significant reduction in gradient communication, as it avoids the need to transmit the entire set of raw parameter differences. Our method offers a highly promising solution for achieving more efficient FL. It greatly enhances communication efficiency, a critical concern in FL systems, ultimately improving the scalability and applicability of FL in real-world applications, particularly for scenarios with limited bandwidth or intermittent connectivity.

The main contributions of this paper are summarized as follows:

- We introduce *Gradient Distillation*, a novel approach of distilling the structural essence of gradients rather than directly compressing them, allowing for the transmission of only the indispensable information for model updates.

- Experimental results demonstrate that gradient distillation reduces communication by orders of magnitude compared to baselines without significant accuracy degradation, enabling highly communication-efficient federated learning.

- Our method provides a new perspective on overcoming communication bottlenecks in federated learning, facilitating the application of federated learning at scale on massively distributed devices with limited bandwidth.

## 2 METHODOLOGY

In this paper, we consider federated learning across $N$ edge devices with heterogeneous bandwidth resources. For each device $i$, it possesses a private local dataset $\mathcal{D}_i = \{(\boldsymbol{x}_j^i, y_j^i)\}_{j=1}^{m_i}$ drawn from a unique distribution $\mathcal{P}_i$ over $\mathcal{X} \times \mathcal{Y}$. Federated learning pursues collaboratively training a global

model without directly accessing private data. To achieve this, edge devices periodically transmit their local model updates to a central server and receive the broadcasted global model. The ultimate objective is to obtain a global model that minimizes the risk across all private datasets, *i.e.*,

$$\arg\min_{w} \mathcal{L}(w) \triangleq \frac{1}{N} \sum_{i=1}^{N} \mathcal{L}_i(w), \tag{1}$$

where $w$ is the parameter of the model, $\mathcal{L}_i(w) = \frac{1}{m} \sum_{j=1}^{m_i} \ell(w; (\boldsymbol{x}_j, y_j))$ denotes the empirical risk with respect to device $i$, and $\ell$ represents the loss function. Our goal is to reduce the communication overhead incurred during this distributed training process.

## 2.1 GRADIENT DISTILLATION

**Motivation.** In each round of FL, the server broadcasts the aggregated global model from the preceding round to the client, while each client performs local training to generate an updated local model. Traditional methods involve transmitting parameter differences between the old and new models. However, this approach maintains a constant data transfer volume irrespective of changes in the numerical value of the parameter difference, as long as the network architecture remains fixed during the FL process. This introduces redundancy in gradient transmission. We claim that this issue arises due to the interdependence between model parameters and network architecture. Since network parameters and structure are intertwined, even slight parameter adjustments necessitate the retransmission of entire models. To counteract this, we propose decoupling model parameters from network architecture to transmit only the essential information required for synchronization.

Specifically, we approximate the gradient disparities between models into a synthetic tensor sequence, employing a distillation-inspired concept. The recipient (*i.e.*, the server side) can then reconstruct the intended model update of the sender (*i.e.*, the client side) by taking a single descent step on these ordered tensors in conjunction with the previous model state. Through this parameter-structure decoupling, our approach transmits only the indispensable information for model updates, bypassing the need for transmitting the whole amount of raw parameter differences. This results in a substantial reduction in gradient communication. We refer to this approach as *gradient distillation*, which will be introduced in detail in the following.

**Gradient Distillation.** For a model with parameters $w$, we define its difference between two periods $t_1$ and $t_2$ as $\Delta\omega(t_1, t_2) = \omega^{(t_1)} - \omega^{(t_2)}$. By approximating $\Delta\omega(t_1, t_2)$, we can update the model parameters at timestamp $t_2$ even when we only have access to the stale model $\omega^{(t_1)}$. To achieve this, we synthesize an ordered tensor sequence $\zeta = \{(\hat{x}_j, \hat{y}_j)\}_{j=1}^m$, which is tailored to approximate the parameter difference using one-step gradient descent on $\omega^{(t_1)}$. Our objective is to *discover the shortest projected path between $\omega^{(t_1)}$ and $\omega^{(t_2)}$*. To synthesize this ordered sequence, we minimize the error between the parameter difference and the sequence gradient on $\omega^{(t_1)}$:

$$\{(\hat{x}_j, \hat{y}_j)\}_{j=1}^m = \arg\min_{\{(x_j, y_j)\}_{j=1}^m} \left\| \mathrm{optim}\left( \sum_{i=1}^m \frac{\partial \ell(\omega; (x_j, y_j))}{\partial \omega} \Big|_{\omega = \omega^{(t_1)}} \right) - \Delta\omega(t_1, t_2) \right\|_2^2, \tag{2}$$

where $m$ represents the solved length of the optimal sample sequence, and $\mathrm{optim}(g)$ denotes some operators performed on the gradient $g$ involved in the optimizer such as SGD or Adam.

Solving this optimization problem directly poses a significant challenge. Therefore, we adopt a greedy approach to find an approximate solution, illustrated in Fig. 1 (c). The objective is to identify a minimal sequence of synthetic tensors, each associated with labels, that can effectively replicate the true gradient when employed to train the original model. This effectiveness is assessed by evaluating the disparity between the reproduced and actual gradients. Through a step-by-step, iterative process, we incrementally construct the optimal sample sequence. This greedy strategy offers a practical means of working towards the broader objective of obtaining a concise statistical representation of localized model updates.

## 2.2 GRADIENT DISTILLATION BASED FEDERATED LEARNING

We now outline the workflow of the proposed Gradient Distillation based Federated Learning (FedGD) framework, encompassing the following key states: initialization, broadcasting, local training, uploading and global aggregation.

– **Initialization:** The first $M$ rounds serve as the initialization stage, for which the training process adheres to the standard federated learning approach. The central server initiates the process by broadcasting an initial global model $\omega_g^{(0)}$ to all participating devices. Each device then conducts local training on this model, utilizing its individual private data. Upon completion of the local training, the devices upload their respective updated local models. Subsequently, the central server aggregates these models to generate a new global model. After $M$ rounds, the central server possesses the global model with parameter $\omega_g^{(M)}$. For the subsequent training rounds, we introduce the proposed FedGD strategy to reduce the communication burden between the central server and the edge clients, for both the global model broadcasting (downlink communication) and local updates uploading (uplink communication). The workflow is offered in Fig. 2 (b). Specifically, for the $t$-th round, our framework works as follows:

– **Broadcasting:** In this stage, the central server performs gradient distillation between the new global model $\omega_g^{(t)}$ and the broadcasted model $\omega_g^{(t-1)}$ in the last round. The objective is to obtain a compressed datastream $\zeta_g^{(t)}$ for downlink communication, which is a tensor of length $m_g^{(t)}$. To achieve this, the server solves the following optimization problem:

$$\zeta_g^{(t)} = \underset{\{(x_j, y_j)\}_{j=1}^{m_g^{(t)}}}{\arg\min} \left\| \text{optim} \left( \sum_{j=1}^{m_g^{(t)}} \left. \frac{\partial \ell(\omega; (x_j, y_j))}{\partial \omega} \right|_{\omega=\omega_g^{(t-1)}} \right) - \Delta\omega_g(t-1, t) \right\|_2^2, \quad (3)$$

where $m_g^{(t)}$ represents the length of tensor $\zeta_g^{(t)}$, and $\Delta\omega_g(t-1, t)$ denotes the difference between $\omega_g^{(t)}$ and $\omega_g^{(t-1)}$. Instead of broadcasting the parameter difference, the server broadcasts $\zeta_g^{(t)} = \{(\hat{x}_j, \hat{y}_j)\}_{j=1}^{m}$ to all participating devices, which significantly reduces the downlink burden.

– **Local Training:** Upon receiving the broadcasted tensor $\zeta_g^{(t)}$, each edge device recovers the intended global model $\omega_g^{(t)}$ using the global model $\omega_g^{(t-1)}$ from the last round. This recovery process is achieved through one-step gradient descent on $\zeta_g^{(t)}$:

$$\omega_g^{(t)} = \omega_g^{(t-1)} - \text{optim} \left( \sum_{(\hat{x}, \hat{y}) \in \zeta_g^{(t)}} \left. \frac{\partial \ell(\omega; (\hat{x}, \hat{y}))}{\partial \omega} \right|_{\omega=\omega_g^{(t-1)}} \right). \quad (4)$$

Furthermore, for each edge device $i$, started from the model $\omega_g^{(t)}$, it performs local training on its private dataset for $K$ iterations to update its parameters as $\omega_i^{(t,k)}$:

$$\omega_i^{(t,k)} \leftarrow \omega_i^{(t,k-1)} - \gamma \sum_{(x,y) \in \mathcal{B}_i} \left. \frac{\partial \ell(\omega; (x, y))}{\partial \omega} \right|_{\omega=\omega_i^{(t,k-1)}} \quad \text{for } k \in [K], \quad (5)$$

where $\omega_i^{(t,0)} = \omega_g^{(t)}$, $\omega_i^{(t)} = \omega_i^{(t,K)}$, and $\mathcal{B}_i$ is the $i$-th random batch drawn from $\mathcal{D}_i$.

– **Uploading:** After local training, each device $i$ executes gradient distillation between the global model $\omega_g^{(t)}$ and the updated local model $\omega_i^{(t)}$:

$$\zeta_i^{(t)} = \underset{\{(x_j, y_j)\}_{j=1}^{m_i^t}}{\arg\min} \left\| \text{optim} \left( \sum_{j=1}^{m_g^{(t)}} \left. \frac{\partial \ell(\omega; (x_j, y_j))}{\partial \omega} \right|_{\omega=\omega_g^{(t)}} \right) - \left( \omega_g^{(t)} - \omega_i^{(t)} \right) \right\|_2^2, \quad (6)$$

where $m_i^t$ denotes the length of tensor $\zeta_i^{(t)}$. The value of $m_i^{(t)}$ is related to the difference between $\omega_i^{(t)}$ and $\omega_g^{(t)}$, which varies in each round. Each device $i$ uploads the synthetic samples $\zeta_i^{(t)}$ to the server, which significantly reduces the uplink communication burden.

– **Global Aggregation:**

Upon receiving the uploaded tensor $\zeta_i^{(t)}$, the central server performs backpropagation on the model $\omega_g^{(t)}$ with data $\zeta_i^{(t)}$ to recover the intended updated local model $\omega_i^{(t)}$ for each device $i$. This is achieved by addressing the following optimization problem:

$$\omega_i^{(t)} = \omega_g^{(t)} - \text{optim} \left( \sum_{(\hat{x}, \hat{y}) \in \zeta_i^{(t)}} \left. \frac{\partial \ell(\omega; (\hat{x}, \hat{y}))}{\partial \omega} \right|_{\omega=\omega_g^{(t)}} \right). \quad (7)$$

---

**Algorithm 1:** Gradient Distillation based Communication-efficient Federated Learning

---

**Input:** $N$ edge devices with private datasets $\{\mathcal{D}_i\}_{i=1}^N$, communication round number $T$, learning rate $\eta$,
    initial rounds $M$, local update number $K$, batchsize $B$.
**Output:** FL-trained global model $\omega_g^T$.
**Server Executes:**
   | **Initialization:** Following the standard FedAvg workflow. After $M$ rounds, the central server has
     $\omega_g^{(M)}$ with $\omega_g^{(M-1)}$, and the device has $\omega_g^{(M-1)}$
   **for** each communication round $t = M, \ldots, T$ **do**
      | Obtain $\zeta_g^{(t)}$ by performing gradient distillation between $\omega_g^{(t-1)}$ and $\omega_g^{(t)}$ with Eq. (3)
      **for** each device $i = 1, 2, \ldots, N$ **in parallel do**
        | Broadcasting $\zeta_g^{(t)}$ to device $i$
        $\zeta_i^{(t)} \leftarrow$ **Device Executes** $(i, \zeta_g^{(t)})$
        Reconstruct the local model $\omega_i^{(t)}$ by one-step gradient descent on $\zeta_t^{(i)}$ with Eq. (7)
      **end**
      $\omega_g^{(t+1)} \leftarrow \frac{1}{N} \sum \omega_i^{(t)}$
   **end**
**end**
**Device Executes** $(i, \zeta_g^{(t)})$**:**
   | Reconstruct the global model $\omega_g^{(t)}$ by one-step gradient descent on $\zeta_g^{(t)}$ with Eq. (4)
   Update the local model as $\omega_i^{(t)}$ by local training on $\mathcal{D}_i$ with Eq. (5)
   Obtain $\zeta_i^{(t)}$ by performing gradient distillation between $\omega_g^{(t)}$ and $\omega_i^{(t)}$ with Eq. (6)
   Return $\zeta_i^{(t)}$
**end**

---

The server then aggregates the reconstructed local models $\omega_i^{(t)}$ from all selected devices to obtain an updated global model $\omega_g^{(t+1)}$:

$$\omega_g^{(t+1)} = \frac{1}{N} \sum_{i=1}^N \omega_i^{(t)}. \tag{8}$$

In essence, the compact tensor sequence $\zeta_i^{(t)}$ allows the server to efficiently recreate device $i$'s full model update through one forward pass, instead of directly transmitting the high-dimensional model weights $\omega_i^{(t)}$, while preserving accurate information to synchronize the global and local models.

**Remark.** As described above, our approach involves performing gradient distillation at both the central server and edge devices, which introduces slightly higher computational demands for these components. It is important to highlight that, in comparison to conventional FedAvg, our method does not incur longer overall training time. On the contrary, it actually leads to much shorter training times per round. This improvement is attributed to the fact that gradient distillation substantially reduces the amount of data that needs to be transmitted, consequently reducing the time required for transmission. This has been empirically verified in experiment section, as shown in Table 3. This efficiency gain is a significant advantage of our approach, as it allows for quicker model updates. It is worth noting that communication resources are typically much more constrained than computation resources. Therefore, the approach of reducing the communication burden by slightly increasing the computation burden is highly practical for many real-world applications.

## 2.3 PARALLEL VERSION

To further mitigate potential efficiency losses due to the extra computational steps, inspired by parallel federated learning framework (Zhang et al., 2023), we concurrently conduct server-side model aggregation while allowing for device-level local training. By overlapping aggregation stage with local update stage, our solution maintains substantial communication savings without compromising computational throughput, as indicated in Fig. 2 (c). Specifically, for the $t$-th round, the parallel version of our framework (Parallel FedGD) works as follows:

**– Client Side.**

Table 1: Test accuracy (%) and data transmission size reduction ratios of FedGD and baseline methods on CIFAR-10 using different networks.

| Model | Method | Avg. data transmission | | Min. data transmission | | Accuracy (%) |
|---|---|---|---|---|---|---|
| | | Volume (MB) | Speedup | Volume (MB) | Speedup | |
| MobileNet | FedAvg | 4.21 | $1\times$ | 4.21 | $1\times$ | 82.48 |
| | Top-k (Aji & Heafield, 2017) | 2.11 | $2\times$ | 2.11 | $2\times$ | 78.85 |
| | FedPAQ (Reisizadeh et al., 2020) | 1.05 | $4\times$ | 1.05 | $4\times$ | 79.77 |
| | DAdaQ (Hönig et al., 2022) | 1.92 | $2.19\times$ | 1.05 | $4\times$ | 80.68 |
| | AdaGQ (Liu et al., 2023) | 1.57 | $2.68\times$ | 1.05 | $4\times$ | 80.22 |
| | FedGD | 0.0328 | $128\times$ | 0.0061 | $690\times$ | 82.19 |
| | Parallel FedGD | 0.0472 | $89\times$ | 0.0061 | $690\times$ | 82.08 |
| ShuffleNet | FedAvg | 5.42 | $1\times$ | 5.42 | $1\times$ | 83.15 |
| | Top-k (Aji & Heafield, 2017) | 2.71 | $2\times$ | 2.71 | $2\times$ | 79.07 |
| | FedPAQ (Reisizadeh et al., 2020) | 1.36 | $4\times$ | 1.36 | $4\times$ | 80.49 |
| | DAdaQ (Hönig et al., 2022) | 2.64 | $2.05\times$ | 1.36 | $4\times$ | 81.87 |
| | AdaGQ (Liu et al., 2023) | 1.98 | $2.74\times$ | 1.36 | $4\times$ | 81.62 |
| | FedGD | 0.0421 | $129\times$ | 0.0031 | $1765\times$ | 82.98 |
| | Parallel FedGD | 0.0571 | $95\times$ | 0.0031 | $1765\times$ | 82.74 |
| ResNet-18 | FedAvg | 11.69 | $1\times$ | 11.69 | $1\times$ | 85.31 |
| | Top-k (Aji & Heafield, 2017) | 5.85 | $2\times$ | 5.85 | $2\times$ | 80.14 |
| | FedPAQ (Reisizadeh et al., 2020) | 2.92 | $4\times$ | 2.92 | $4\times$ | 83.39 |
| | DAdaQ (Hönig et al., 2022) | 4.97 | $2.35\times$ | 2.92 | $4\times$ | 82.47 |
| | AdaGQ Liu et al. (2023) | 4.09 | $2.86\times$ | 2.92 | $4\times$ | 82.09 |
| | FedGD | 0.0369 | $317\times$ | 0.0092 | $1268\times$ | 85.11 |
| | Parallel FedGD | 0.0508 | $230\times$ | 0.0154 | $759\times$ | 85.03 |

• **Local Training:** Each device $i$ receives the broadcasted gradient distillation tensor $\zeta_g^{(t-1)}$ from the server, and reconstructs the global model $\omega_g^{(t-1)}$ based on the previous model $\omega_g^{(t-2)}$:

$$\omega_g^{(t-1)} = \omega_g^{(t-2)} - \text{optim}\left(\sum_{(\hat{x},\hat{y})\in\zeta_g^{(t-1)}} \left.\frac{\partial\ell(\omega;(\hat{x},\hat{y}))}{\partial\omega}\right|_{\omega=\omega_g^{(t-2)}}\right). \qquad (9)$$

Device $i$ then performs local training to get the updated local model $\omega_i^{(t)}$ according to Eq. (5).

• **Uploading:** After completing the local training, device $i$ performs gradient distillation to get $\zeta_i^{(t)}$ based on $\omega_i^{(t)}$ and $\omega_g^{(t-1)}$ according to Eq. (6). $\zeta_i^{(t)}$ is then uploaded to the central server.

**– Server Side.**

• **Broadcasting:** During local training and gradient distillation on edge device $i$ to obtain the local model $\omega_i^{(t)}$ and distilled tensor $\zeta_i^{(t)}$, the central server performs global aggregation in parallel to derive the updated global model $\omega_g^{(t)}$ and its distilled form $\zeta_g^{(t)}$. By leveraging parallel optimization, when the server receives all distilled local gradients $\zeta_i^{(t)}$, it will already have finalized the updated global gradient $\zeta_g^{(t)}$ through concurrent aggregation. The server can then immediately broadcast $\zeta_g^{(t)}$ to the edge devices to start the next round of federated optimization.

• **Global Aggregation:** While the broadcast stage is performing, the central server reconstructing $\omega_i^{(t)}$ by:

$$\omega_i^{(t)} = \omega_g^{(t-1)} - \text{optim}\left(\sum_{(\hat{x},\hat{y})\in\zeta_g^{(t)}} \left.\frac{\partial\ell(\omega;(\hat{x},\hat{y}))}{\partial\omega}\right|_{\omega=\omega_g^{(t-1)}}\right). \qquad (10)$$

Based on $\omega_i^{(t)}$, the central server can get the new global model: $\omega_g^{(t+1)} = \frac{1}{N}\sum_{i=1}^{N}\omega_i^{(t)}$.

It then performs gradient distillation between the two latest global models:

$$\zeta_g^{(t+1)} = \underset{\{(x_j,y_j)\}_{j=1}^{m_g^{t+1}}}{\arg\min}\left\|\text{optim}\left(\sum_{j=1}^{m_g^{t+1}} \left.\frac{\partial\ell(\omega;(x_j,y_j))}{\partial\omega}\right|_{\omega=\omega_g^{(t)}}\right) - \Delta\omega_g(t,t+1)\right\|_2^2. \qquad (11)$$

As illustrated in Fig. 2, when the central server performs global aggregation and distillation to derive $\omega_g^{(t)}$ and $\zeta_g^{(t)}$, edge devices simultaneously conduct local training and gradient distillation to obtain

Table 2: Test accuracy (%) and data transmission size reduction ratios of FedGD and baseline methods on CIFAR-100 and PathMNIST using MobileNet.

| Model | Method | Avg. data transmission | | Min. data transmission | | Accuracy (%) |
|---|---|---|---|---|---|---|
| | | Volume (MB) | Speedup | Volume (MB) | Speedup | |
| CIFAR-100 | FedAvg | 11.69 | 1× | 11.69 | 1× | 60.17 |
| | Top-k (Aji & Heafield, 2017) | 5.85 | 2× | 5.85 | 2× | 56.26 |
| | FedPAQ (Reisizadeh et al., 2020) | 2.92 | 4× | 2.92 | 4× | 57.38 |
| | DAdaQ (Hönig et al., 2022) | 6.22 | 1.88× | 2.92 | 4× | 58.51 |
| | AdaGQ (Liu et al., 2023) | 5.65 | 2.07× | 2.92 | 4× | 58.37 |
| | FedGD | 0.0989 | 118× | 0.0122 | 958× | 59.84 |
| PathMNIST | FedAvg | 11.69 | 1× | 11.69 | 1× | 87.81 |
| | Top-k (Aji & Heafield, 2017) | 5.85 | 2 × | 5.85 | 2× | 83.20 |
| | FedPAQ (Reisizadeh et al., 2020) | 2.92 | 4 × | 2.92 | 4 × | 84.74 |
| | DAdaQ (Hönig et al., 2022) | 4.97 | 2.36 × | 2.92 | 4 × | 86.17 |
| | AdaGQ Liu et al. (2023) | 4.09 | 2.87× | 2.92 | 4× | 85.92 |
| | FedGD | 0.0345 | 339 × | 0.0614 | 1904 × | 87.54 |

$\omega_i^{(t)}$ and $\zeta_i^{(t)}$. This parallel execution scheme, with the server and devices simultaneously performing their respective operations, significantly improves the computational efficiency of FedGD.

## 3 EXPERIMENTS

### 3.1 EXPERIMENTAL SETUP

**Baselines.** We compare our proposed FedGD approach with five baseline methods: (1) FedAvg, (2) Top-k (Aji & Heafield, 2017), (3) FedPAQ (Reisizadeh et al., 2020), (4) DAdaQ (Hönig et al., 2022), and (5) AdaGQ (Liu et al., 2023).

**Datasets.** Experiments are conducted on three benchmark datasets: CIFAR-10 (Krizhevsky et al., 2009) , CIFAR-100, and a medical image dataset PathMNIST (Yang et al., 2023). PathMNIST is a dataset for predicting survival from colorectal cancer histology slides. It contains 100,000 images in the training set and 7,180 images in the test set. The image size is $3 \times 28 \times 28$. For CIFAR-10, we evaluate the generalization of our approach to different network architectures by testing with MobileNet (Howard et al., 2017), ShuffleNet (Zhang et al., 2018), and ResNet-18 (He et al., 2016). To validate performance across datasets, we use MobileNet for experiments on CIFAR-100 and PathMNIST. All baselines use the same network architecture as FedGD for a fair comparison.

**Implementation Details.** We implement FedGD as well as the baseline methods using the PyTorch framework. The SGD optimizer with a learning rate of 0.01 is used for all approaches. Unless otherwise stated, the batch size is 64 and the number of local epochs per round is set to 3. The training process spans 150 communication rounds, with gradient distillation initiated from the 31st round onwards. Following the commonly used simulation setting (Liu et al., 2023; Hönig et al., 2022), in our experiments, we simulate 50 virtual devices, and set the bandwidth to 50 Mbps.

### 3.2 PERFORMANCE EVALUATION

Table 1 shows the test accuracy, average and minimum data transmission size reduction ratios of FedGD and the baseline methods in CIFAR-10 using three widely used network architectures. FedAvg serves as the comparison benchmark. Among the compared methods, Top-k and FedPAQ, which do not adopt adaptive schemes, have equal average and minimum compression ratios, and lower accuracy than FedAvg (by 3.63% and 2.71% for MobileNet). DAdaQ and AdaGQ apply different adaptive quantization schemes based on time and gradient norms. AdaGQ saves 2.68× in communication compared to 2.19× for DAdaQ, but with a 0.46% lower precision. Our method, FedGD achieves higher precision than all baselines at 82. 08%, only 0. 4% lower than the uncompressed FedAvg. Most notably, it provides average 128× compression, reaching up to 690× in later rounds as the differences in network parameters shrink. Across MobileNet, ShuffleNet, and ResNet-18, FedGD significantly outperforms baselines in balancing training accuracy and communication efficiency. To evaluate the generalizability of our approach across different datasets, we further validate it on CIFAR-100 and PathMNIST in addition to the previous experiments. As shown in Table

Table 3: Results with different batchszie, local update number and Pre-distillation rounds. $T_g$ represent the calculated time for gradient distillation, $T_c$ represents the upload communication time of standard FL, $T_c^*$ represents the time of upload communication using gradient distillation FL.

| Batchszie $B$ | Avg. data volume (MB) | Accuracy (%) | $T_g$ (s) | $T_c^*$ (s) | $T_g + T_c^*$ (s) | $T_c$ (s) | Speedup |
|---|---|---|---|---|---|---|---|
| 32 | $3.28 \times 10^{-2}$ | 85.34 | 8.97 | 0.05 | 9.02 | 18.78 | $2.08\times$ |
| 64 | $3.69 \times 10^{-2}$ | 85.11 | 10.09 | 0.06 | 10.15 | 18.78 | $1.85\times$ |
| 128 | $4.47 \times 10^{-2}$ | 84.97 | 12.22 | 0.07 | 12.29 | 18.78 | $1.53\times$ |
| Local round number $K$ | Avg. data volume (MB) | Accuracy | $T_g$ (s) | $T_c^*$ (s) | $T_g + T_c^*$ (s) | $T_c$ (s) | Speedup |
| 1 | $2.08 \times 10^{-2}$ | 85.79 | 5.69 | 0.03 | 5.72 | 18.78 | $3.28\times$ |
| 2 | $2.73 \times 10^{-2}$ | 85.42 | 7.46 | 0.04 | 7.50 | 18.78 | $2.5\times$ |
| 3 | $3.69 \times 10^{-2}$ | 85.11 | 10.09 | 0.06 | 10.15 | 18.78 | $1.85\times$ |
| Pre-distillation rounds $M$ | Avg. data volume (MB) | Accuracy | $T_g$ (s) | $T_c^*$ (s) | $T_g + T_c^*$ (s) | $T_c$ (s) | Speedup |
| 30 | $3.69 \times 10^{-2}$ | 85.11 | 10.09 | 0.06 | 10.15 | 18.78 | $1.85\times$ |
| 50 | $2.95 \times 10^{-2}$ | 85.17 | 8.07 | 0.05 | 8.12 | 18.78 | $2.31\times$ |
| 80 | $2.12 \times 10^{-2}$ | 85.19 | 5.80 | 0.03 | 5.83 | 18.78 | $3.22\times$ |

2, our method achieved the highest accuracy on both CIFAR-100 and PathMNIST, demonstrating strong performance generally. Specifically on CIFAR-100, our results exceeded the best-performing baseline method DAdaQ by 1.33% in accuracy while still providing an average $118\times$ reduction in communication costs. On PathMNIST as well, we attained superior test precision compared to alternative techniques. This confirms our gradient distillation method is capable of adapting to diverse data distributions without sacrificing predictive power or federated learning objectives.

In summary, FedGD achieves state-of-the-art compression ratios while maintaining high model performance, validating the effectiveness of our proposed gradient distillation approach.

### 3.3 ABLATION ANALYSIS ON HYPERPARAMETERS

Table 3 displays the compression ratios achieved by our method for varying batch sizes (32, 64, 128), round numbers of local updates (1, 2, 3), and pre-distillation rounds (30, 50, 80). It is evident that lower batch sizes and fewer local updates result in higher compression, as they entail smaller model parameter differences between rounds, containing less information for compression. Notably, compared to a batch size of 128, using a batch size of 32 further reduces communication by $1.36\times$. Moreover, increasing the number of pre-distillation rounds (*i.e.*, applying compression in later training stages) leads to a reduction in average uploaded parameters, as differences diminish over time. When $M = 80$ rounds, only an average of $2.12 \times 10^{-2}$ MB parameters are uploaded.

Additionally, we provide the computation time $T_g$ for gradient distillation, the communication time $T_c^*$ for uploading tensors generated by gradient distillation, and the communication time $T_c$ for uploading model differences. The combined time for $T_g$ and $T_c^*$ is significantly smaller than $T_c$, indicating that our method substantially shortens the training time per round. This improvement is attributed to the fact that gradient distillation significantly reduces the amount of data that needs to be transmitted, consequently reducing the time required for transmission.

## 4 CONCLUSION

In this study, we introduced gradient distillation-based communication-efficient federated learning (FedGD), a novel approach where devices and the server synthesize tensor sequences to represent model updates, rather than transmitting raw model differences. Our key innovation lies in distilling the structural essence of gradients, as opposed to directly compressing them. This enables the transmission of only essential information for synchronization, bypassing the need to transmit the entire set of raw parameter differences. Experimental results demonstrate that FedGD substantially reduces communication overhead without sacrificing accuracy significantly. This highlights its potential to scale privacy-preserving distributed training across edge networks by leveraging model-specific representations.

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
