# OpenReview forum: "Communication-Efficient Federated Learning via Gradient Distillation"
_ICLR.cc/2024/Conference — ICLR 2024 Conference Withdrawn Submission_

### Official Review · Reviewer_tMVR · 2023-10-29

**Soundness:** 2 fair
**Presentation:** 2 fair
**Contribution:** 2 fair
**Rating:** 3
**Confidence:** 4

**Summary:**

This paper proposes a new approach, termed gradient distillation, for reducing the bits communicated between the server and clients in a federated learning setup. The key idea of the gradient distillation is to learn a synthetic dataset on both the server side and the client side, and the model's gradient on this synthetic dataset can match the global/local difference. Such synthetic dataset is obtained by solving an optimization problem where the objective is the $\ell_2$ difference of ground truth model difference and model's gradient on the synthetic dataset. Experiments results show that the total volume of such learned synthetic dataset can be significantly smaller than the original model difference, and by only transmitting those synthetic datasets, the algorithm can still achieve similar performance to the original FedAvg algorithm.

**Strengths:**

The strength is clear, as demonstrated in the experiment part, the data transmission volume can be reduced significantly compared to the existing communication-efficient FL algorithm on the two tested datasets. This work reminds me of the work on dataset distillation [1], which shares a similar motivation for learning a compressed dataset. The experiment results are consistent with those in [1], as both this work and [1] show that the training dataset can be compressed significantly without losing the performance too much.


reference: [1] https://arxiv.org/pdf/1811.10959.pdf

**Weaknesses:**

Despite that it is truly surprising to see that, as shown by this work, the local dataset can be compressed up to an extreme level. I do not think this work is solid enough for publishing, below is my justification.

1. Lack of theoretical justification

The idea of this work, albeit interesting, does not have any theoretical analysis of why and when the algorithm can work. Without such analysis, it can hardly be convincing that this algorithm can perform well under any circumstances. So far I can only tell that this algorithm can work with the two selected datasets, and am not sure whether these datasets are cherry-picked.

To make this paper solid, I would suggest the author either test this algorithm on even more datasets and more network architectures because I think the experiment with only two selected datasets is far from solid for an entirely empirical work, or the author could provide a theoretical explanation for the proposed algorithm.

For the theoretical results, I have some suggestions for the author. Firstly, I can tell that as long as the optimal $\ell_2$ error of the optimization problem (2) is small enough at every step, the algorithm can certainly have a convergence guarantee. On the other hand, it is intuitive to see that as long as $m$ is sufficiently large, the error of (2) can be arbitrarily small, but $m$ is related to the transmission volume. With this observation, a natural theoretical question is, given small error bound $\epsilon$, what is the least value of $m$ such that the optimal error of (2) is smaller than $\epsilon$? I believe that this manuscript can be more solid if it includes some theoretical exploration on this point.

2. Quadratically scaling computational cost

I am also concerned with the computation complexity of the optimization problem (2). It seems that the complexity is quadratic of the numbers of the network parameters, because it needs to compute the hessian of the network parameters during backpropagation. I think maybe this is why the experiments part only includes small networks like Mobilenet, ShuffleNet and ResNet-18. I would suggest the author to discuss more on this point.

**Questions:**

I am not certain how the optimization problem (2) is solved,  the author only briefly described that  adopt a greedy strategy in the manuscript. Based on my own knowledge, I guess the method is to solve for only one data tensor at each time and add it to the total dataset until the optimization error is lower than a certain value. Am I correct?

---

### Official Review · Reviewer_avnc · 2023-10-31

**Soundness:** 2 fair
**Presentation:** 3 good
**Contribution:** 2 fair
**Rating:** 3
**Confidence:** 4

**Summary:**

Communication efficiency is a central issue in federated learning (FL). Parameter updates are transmitted frequently during federated training to avoid direct private data sharing. As deep learning models grow larger and larger, huge communication costs may prevent these models from being trained in FL settings due to limited bandwidth of edge devices. To improve communication efficiency, this paper introduced gradient distillation that approximates gradient disparities into a synthetic tensor sequence, allowing the recipient to reconstruct the sender’s intended model update. Experimental results demonstrated that gradient distillation reduces communication by orders of magnitude compared to baselines without significant accuracy degradation.

**Strengths:**

1. The paper provided an interesting approach to reduce communication cost in FL, by distilling gradient information into synthetic data samples, and transmitting the low-cost data samples to approximately recover gradients. Dataset distillation has been an active topic and related techniques may be applied to FL via the proposed framework.

2. The paper well motivated the gradient distillation approach and explained the methodology.

**Weaknesses:**

1. While the paper is motivated by efficiently training large models in FL, The proposed method may face with scalability issues as well. When the dataset is complex and the local training process goes longer, solving the optimization problem in Equation (2) (even approximately) can be challenging. In other words, it would be hard to use a small ratio of synthetic samples to represent a long training trajectory without significant accuracy loss.

2. The proposed method introduced an additional optimization process for gradient distillation on local devices, the computation cost of which should not be underestimated. In the last paragraph of Section 2.2, the paper remarked that the proposed method leads to much shorter training times per round, and that communication resources are typically much more constrained than computation resources. However, the statements are not supported by experimental results or previous studies. In Table 3 the paper showed communication time, given a fixed bandwidth, and computation time for gradient distillation, but the computation time of local training is missing. Also the computation capacity of devices to run the experiments is not provided. Note that it is unfair to use server-level devices to present computation time, and mobile-level bandwidth to present communication time.

3. Some key experimental information is missing or confusing. See Questions for details.

**Questions:**

1. What is the specific procedure of the greedy approach mentioned in the last paragraph of Section 2.1?

2. Can the method be applied to optimizers like Adam? How would the optim(.) operation affect the performance of gradient distillation?

3. In experiments, how is the number m of synthetic samples determined?

4. In Table 1 and Table 2, the accuracies are high, given that there are 50 clients. How are the data distributed among clients? Is the data distribution IID or non-IID? Moreover, how is the average speedup factor computed? From the order of speedup (~100x) one can see it does not take the first M standard rounds into consideration. As 30 out of 150 rounds are standard FL training, the overall average speedup should be considerably smaller.

5. In Section 3.1 the paper mentioned “the number of local epochs per round is set to 3”. Here, does “epoch” refer to the number of update steps, as K in Equation (5), or the number of passes over the local dataset? Note that results in Table 3 used K.

6. What is the computation capacity of devices for running the experiments?

7. What is the speedup for the total training process, including all computation and communication steps? This again depends on consistent assumptions of computation power and communication bandwidth.

---

### Official Review · Reviewer_RHbi · 2023-11-07

**Soundness:** 2 fair
**Presentation:** 3 good
**Contribution:** 2 fair
**Rating:** 3
**Confidence:** 5

**Summary:**

This paper proposes an idea called gradient distillation in the context of federated learning, which decouples model parameters from the network architecture, enabling the transmission of only essential information needed for synchronization during the federated training.

**Strengths:**

The paper is mostly well-written. Although I found some hyperbole claims. The graphics is good, but they can be compact, especially Figure 2.

**Weaknesses:**

Please see below.

**Questions:**

1.	In my understanding, distributed mean estimation methods reduce the normalized mean square error between each client’s approximated vector to the estimated mean; for example, please check [1]. Can the authors call their Gradient Distillation (GD) as a form of distributed mean estimation which in their own words “minimize the error between the parameter difference and the sequence gradient on $\omega^{(t_1)}$? I think they can make connection and if that is so, any distributed mean estimation method is suitable for communication compression in distributed learning ---- FL is a special case of distributed learning. So, in that case, the authors should tone down their cliam of the “groundbreaking concept.” I also sincerely suggest the authors should make sufficient connection between these two ideas.

2.	To discuss about communication efficiency in federated learning, the authors discussed about two approaches. They are correct—both approaches have their limitations. To mitigate the shortcomings of the first approach, personalized FL models aimed at balancing the trade-off between the traditional global model and the local models that could be trained by individual devices using their private data only. Moreover, there are methods such as loopless gradient descent (L2GD) in [2] proposed a probabilistic gradient descent algorithm to reduce the challenges of the first approach. Furthermore, [3] uses compression techniques on top of L2GD’s probabilistic communication at both central server and the participating local devices. In training similar DNN models as in this work (ResNet-18 and MobileNet on CIFAR-10) [3] to obtain the same global Top-1 test accuracy, compressed L2GD reduces renders approximately $10^4$ times improvement compared to FedAvg in compressing the data-volume. Since this is a highly empirical paper, I highly encourage the authors to provide discussion and benchmarking against personalized FL models before claiming supremacy of their proposed method. The authors should abundantly discuss about these methods.

3.	The authors mentioned “ultimately improving the scalability and applicability of FL in real-world applications, particularly for scenarios with limited bandwidth or intermittent connectivity.” However, I fail to see any limited bandwidth or intermittent connectivity in their experiments. How do the authors claim this if the network bandwidth is set constant to 50 Mbps?

4.	Moreover, I also do not understand, how do the authors emulate the FL client-server architecture. Do the authors use a data-center set-up? Do the authors use multiple server grade machines and partition the GPUs to “synthetically” create a FL environment? Can they kindly elaborate? How do the authors simulate 50 virtual devices? Do all the devices participate in training in each round?

5.	Related to Question 1: Gradient Knowledge Distillation (GKD) is not a brand-new idea. To incorporate the gradient alignment objective into the distillation process, [5] aligns the change of the model around the inputs by introducing an extra objective on gradient consistency between the student and the teacher. Furthermore, [4] uses ensemble knowledge distillation as a multi-objective optimization problem so that one can determine a better optimization direction for the training of student network. In my opinion, these are not exactly mean estimation process, but similar. More strikingly, they are very similar to the ideas the authors proposed in this manuscript. Please correct me if I am wrong.

6.	I did not understand the authors’ claim of adopting a greedy approach to find an approximate solution to (2). Can the authors please elaborate more on this?

7.	What does the authors mean by “synthesizing an ordered tensor…”. Is that the input points? If so, please clarify and use it consistently as mentioned in the beginning of Section 2. What is the notion of having a parallel method? Is not it straightforward?

8. In Table 3, what model and dataset are used? Where are the time comparisons of the other baseline methods?

**References**

1.	Konečný J, Richtárik P. Randomized distributed mean estimation: Accuracy vs. communication. 2018.
2.	F. Hanzely and P. Richtárik, “Federated learning of a mixture of global and local models,” 2020.
3.	Bergou, E.H., et al., Personalized Federated Learning with Communication Compression. In TMLR 2023.
4.	Du et al., Agree to Disagree: Adaptive Ensemble Knowledge Distillation in Gradient Space, NeuRIPS 2020.
5.	Wang et al., Gradient Knowledge Distillation for Pre-trained Language Models, NeuRIPS Workshop 2022.